# Health-related educational inequalities in paid employment across 26 European countries in 2005–2014: repeated cross-sectional study

Jolinda L D Schram,[1] Merel Schuring,[1] Karen M Oude Hengel,[1,2] Alex Burdorf[1]

¹Department of Public Health, Erasmus University Medical Center, Rotterdam, The Netherlands
²Netherlands Organization for Applied Scientific Research TNO, Leiden, The Netherlands

**Correspondence to**
Dr Merel Schuring;
m.schuring@erasmusmc.nl

## ABSTRACT

**Objective** The study investigates the trends in health-related inequalities in paid employment among men and women in different educational groups in 26 countries in 5 European regions.

**Design** Individual-level analysis of repeated cross-sectional annual data (2005–2014) from the EU Statistics on Income and Living Conditions.

**Setting** 26 European countries in 5 European regions.

**Participants** 1 844 915 individuals aged 30–59 years were selected with information on work status, chronic illness, educational background, age and gender.

**Outcome measures** Absolute differences were expressed by absolute differences in proportion in paid employment between participants with and without a chronic illness, using linear regression. Relative differences were expressed by prevalence ratios in paid employment, using a Cox proportional hazard model. Linear regression was used to examine the trends of inequalities.

**Results** Participants with a chronic illness had consistently lower labour force participation than those without illnesses. Educational inequalities were substantial with absolute differences larger within lower educated (men 21%–35%, women 10%–31%) than within higher educated (men 5%–13%, women 6%–16%). Relative differences showed that low-educated men with a chronic illness were 1.4–1.9 times (women 1.3–1.8 times) more likely to be out of paid employment than low-educated persons without a chronic illness, whereas this was 1.1–1.2 among high-educated men and women. In the Nordic, Anglo-Saxon and Eastern regions, these health-related educational inequalities in paid employment were more pronounced than in the Continental and Southern region. For most regions, absolute health-related educational inequalities in paid employment were generally constant, whereas relative inequalities increased, especially among low-educated persons.

**Conclusions** Men and women with a chronic illness have considerable less access to the labour market than their healthy colleagues, especially among lower educated persons. This exclusion from paid employment will increase health inequalities.

## Strengths and limitations of this study

► This is the first study to present information on trends in health-related educational inequalities in paid employment among persons in different European regions

► A strength of the current study is the use of cross-sectional data from a large number of European countries and large number of individuals for which the outcomes are harmonised (1 844 915 individuals in 26 countries).

► The effects of having a chronic illness on being employed (selection) and unemployment causing ill health such as chronic illness (causation) cannot be distinguished by using cross-sectional data.

► Although the outcomes of EU Statistics on Income and Living Conditions are harmonised, comparisons in self-rated health measures between different cultures need to be made with caution.

number of elderly people as a share of those of working age, will rise steadily from 29.4 in 2016 to 50.3 by 2050 in Europe.[1 2] In response, many countries have raised the statutory retirement age.[3] The challenge is ensuring that those of working age are actually participating in the labour force.[4] Health-related exclusion from the labour market is present in many countries,[5 6] whereby workers with a chronic illness are at increased risk of becoming disabled or unemployed. Leaving paid employment may further impact health negatively.[7]

Studies have shown that the presence of a chronic disease is an important barrier for entering and maintaining paid employment.[5] Educational inequalities in morbidity, such as chronic health conditions, are well documented across European countries.[8] Some studies have suggested that the effect of chronic illness on maintaining paid employment differs by socioeconomic position. Bartley and Owen reported that workers with a chronic disease in lower socioeconomic position had more

## INTRODUCTION

Life expectancy continues to increase and will ensure that the old-age dependency ratio, the

difficulties staying in paid employment than those with a higher socioeconomic position during times of economic crisis.[9] In Sweden and the UK, labour market attachment for those with a chronic illness is reduced, with the effect being substantially stronger in the UK than in Sweden.[10 11]

The differential effect of socioeconomic position on the association between chronic illness and paid employment may be modified by national social and economic conditions, such as labour market policies.[12] European countries can be categorised in five regions: the Nordic, Continental, Anglo-Saxon, Southern and Eastern regions.[13 14] This typology reflects general differences in employment protection and income security. Non-employment rates for persons with a longstanding illness as well as a low educational level were found particularly high in the Anglo-Saxon and Eastern regions, compared with the Nordic region.[15] The poor position of disabled people was suggested to be partly counteracted by employment protection,[16] although inequalities in health in itself were not consistently smaller in the Nordic region.[17]

Few studies have documented the trends over time in socioeconomic differences in labour force participation between persons with and without a chronic illness. A repeated cross-sectional study over 25 years in Britain showed that within the lower occupational class, persons with a limiting illness had greater and more rapidly widening gaps in labour force participation than in the higher occupational class.[18] A comparison between the Nordic countries and the UK showed that, since the mid-1990s, there has been a deterioration in paid employment among persons with limiting longstanding illness with a low education, even when the economy was doing well.[19]

Limited information exists on trends in health-related educational inequalities in paid employment among persons in different European regions, since most studies have focused on a single country or compared a selected group of countries. Therefore, the aims of this paper are (1) to examine absolute and relative differences in paid employment between persons with and without a chronic illness within each educational group in five European regions and (2) to describe trends in socioeconomic differences in paid employment between people with and without a chronic illness within European countries.

## METHODS
### Data
A repeated cross-sectional study was conducted with the European Union Statistics on Income and Living Conditions (EU-SILC) survey. The EU-SILC is a survey questionnaire designed to provide comparable information across the EU, and cross-sectional data from 2005 to 2014 were used from 26 countries that participated in EU-SILC since 2005 (Denmark, Finland, Iceland, Norway, Sweden, Austria, Belgium, Germany, France, Luxembourg, the Netherlands, Cyprus, Greece, Italy, Portugal, Spain, Ireland, the UK, Czech Republic, Estonia, Hungary,

Latvia, Lithuania, Poland, Slovakia and Slovenia). Detailed information on the EU-SILC study design can be found elsewhere.[20]

Respondents were selected with information on work status, chronic illness, educational background, age and gender. After excluding individuals who were younger than 30 or older than 59 (around 50% of the sample) and deleting observations with relevant missing information (almost all due to no information on chronic disease, 16% of the remaining sample), our study comprised 179 724 individuals for 2014. For the trend analysis 1 844 915 individuals were selected aged 30–59 years between 2005 and 2014. The age limit of 59 years was applied because a number of countries have an official retirement age of 60, thereby greatly reducing the proportion of employed persons after 60 years. The age limit of 30 years was used to exclude potential confounding due to different educational pathways between those with a chronic illness and those without.

### Employment status
Employment status was classified into six mutually exclusive categories: employed (working full or part time, self-employed full or part time), unemployed, economically inactive (fulfilling domestic tasks and care responsibilities and other inactivity), retired, disabled and other (student, in compulsory military community or service). For the main analysis, a dichotomised variable was used: paid employment or outside of paid employment (all other options).

### Chronic illness
The question 'Do you have any longstanding illness or health problem?' was used to distinguish between people with or without a chronic illness.

### Sociodemographic variables
Three sociodemographic variables were used in the study: age, gender and education. Age was categorised into three 10 year age groups: 30–39, 40–49 and 50–59 years. Educational level was recorded as the highest level of education completed by the respondent. We coded education according to the 1997 International Standard Classification of Education and categorised as low (0–2: preprimary, primary and lower secondary education), intermediate (3–4: upper secondary education/postsecondary non-tertiary) and high (5–6: tertiary education).

### Regions
The European countries were categorised into five different regions based on relative comparability of welfare state regimes and geography.[13] The Nordic region included Denmark, Finland, Iceland, Norway and Sweden. The Continental region included Austria, Belgium, Germany, France, Luxembourg and the Netherlands. The Southern region included Cyprus, Greece, Italy, Portugal and Spain. The Anglo-Saxon region included Ireland and the UK. The Eastern region included the Czech Republic, Estonia, Hungary, Latvia, Lithuania, Poland, Slovakia and Slovenia.

## Statistical analysis

The statistical analysis focuses on both absolute and relative differences in labour force participation among those with and without a chronic disease across different educational groups. The absolute difference depicts the reduction of being in paid employment among persons with a chronic illness and is of special interest as it presents the impact on the total population in percentage point employment rate. The relative difference presents the prevalence ratio as a measure of the strength of the association between having a chronic illness and being out of paid employment. Absolute and relative inequalities may differ because levels of labour force participation are generally higher in higher educated groups.

Annual sample weights were used to estimate the prevalence of labour market participation that are representative for the countries included in this study. For each region, the association of having a chronic illness with labour force participation within each educational group was analysed using two different models. A linear regression model was used to calculate absolute differences in labour force participation between persons with and without a chronic illness. Cox regression analysis with a robust estimate was applied to calculate relative differences.

The linear regression model to calculate absolute difference can be written as:

$$y = \alpha + \beta_1 low + \beta_2 intermediate + \beta_3 low_{chronic} + \beta_4 intermediate_{chronic} + \beta_5 high_{chronic} + \beta_6 age$$

where $y$ is a dichotomous variable for being employed; $\alpha$ is an intercept, two dummy variables to compare low and intermediate education with high education and three additional dummy variables to compare those with a chronic illness to those without in each education group. Adjustment for age was done by default. In this binary linear regression analysis, the dependent variable expresses the proportional difference of being in paid employment (within the range of 0.0–1.0), comparing subjects with a chronic disease to subjects without a chronic disease in the same educational group. Thus, the proportional difference equals the absolute difference or gap between proportions of being in paid employment, which is our primary measure of interest.[21]

The Cox proportional hazard model to calculate relative differences can be written as:

$$h(t, x) = h_0(t) \exp\left( \sum_{k=1}^{p} \beta_1 low + \beta_2 intermediate + \beta_3 low_{chronic} + \beta_4 intermediate_{chronic} + \beta_5 high_{chronic} + \beta_6 age \right)$$

where $h(t,…)$ is the likelihood at time $t$ that a person is not employed, two dummy variables for low and intermediate education to compare to high education and three dummy variables to compare those with a chronic illness

in each education group. Adjustment for age was done by default. Cox regression analysis was preferred above a logistic regression analysis, since it will more accurately estimate the prevalence ratio, which is much closer to the relative risk between being in paid employment and having a chronic illness than the ORs in a logistics regression analysis, since the OR will overestimate the risk ratio due to the high prevalence of being in paid employment in the current study.[22 23]

The time period for all persons was set at a constant level (value 1) to estimate the prevalence ratio (PR).[22 24] In all analyses, variables were recoded so a PR above 1 indicates an increased likelihood of being out of paid employment with corresponding 95% CIs. To account for differences in labour market participation between men and women across regions,[25] the associations were stratified by sex and region, and adjusted for age. Significance was determined by p<0.05.

### Trend analysis in the five regions

For determining the trends in absolute and relative health-related inequalities in being in paid employment over 10 years in each region, we used linear regression analyses with the absolute gap and prevalence ratios in each educational group as dependent variable in each survey year. The analysis was stratified by region and gender. The year of data collection was used as the independent variable.

### Health-related educational inequalities in paid employment

To estimate educational inequalities in the relative differences in labour force participation among persons with a chronic illness, a counterfactual scenario was introduced. In this scenario, it is assumed that the relative difference in labour force participation between those with a chronic illness and those without in the lower educational group can be reduced to the relative difference observed in the higher educational group.[26] The estimation of the excess relative difference in paid employment for chronic illness status in lower educated workers is a fair reflection of educational inequalities between low and high-educated persons. The Cox proportional hazard model was used to estimate this excess relative difference by the formula: (bèta for relative difference in the low-educated group—beta for relative difference in the high-educated group)/(bèta of the relative difference in the low-educated group).

The main analysis was performed using Stata statistical software V.14 (StataCorp). The estimates of paid employment per country and the trend analysis were conducted in IBM SPSS statistical software V.22 (SPSS).

### Patient involvement

Not applicable; we have used available datasets from Eurostat for secondary data analysis.

## RESULTS

The study population consisted of 179 724 participants of whom half were women, the largest group was

**Table 1** Characteristics of the EU-SILC study population in 2014 by European region—30–59-year olds

| | | Nordic (DK,FI, SE, NO, IS) | Western (DE, AT, NL, BE, LU, FR) | Anglo-Saxon (IE, UK) | Southern (CY, ES, IT, EL, PT) | Eastern (EE, LT, LV, CZ, SK, PL, HU, SI) |
|---|---|---|---|---|---|---|
| | | n=15 711 (%) | n=41 624 (%) | n=13 995 (%) | n=52 961 (%) | n=55 433 (%) |
| Sex | Female | 50 | 53 | 53 | 52 | 54 |
| Education | High | 46 | 36 | 44 | 26 | 25 |
| | Intermediate | 42 | 47 | 29 | 36 | 64 |
| | Low | 13 | 17 | 28 | 38 | 11 |
| Age (years) | 30–39 | 27 | 27 | 33 | 28 | 29 |
| | 40–49 | 35 | 36 | 35 | 37 | 33 |
| | 50–59 | 38 | 37 | 33 | 34 | 38 |
| Employment status | Employed | 84 | 80 | 74 | 67 | 76 |
| | Unemployed | 5 | 6 | 7 | 16 | 9 |
| | Economically inactive | 3 | 7 | 10 | 11 | 6 |
| | Retired | 0 | 2 | 2 | 3 | 3 |
| | Disabled | 5 | 4 | 6 | 2 | 6 |
| | Other | 2 | 0 | 1 | 0 | 0 |
| Chronic illness | Yes | 33 | 30 | 27 | 21 | 29 |

AT, Austria; BE, Belgium; CY, Cyprus; CZ, Czech Republic; DE, Germany; DK, Denmark; EE, Estonia; EL, Greece; ES, Spain; EU-SILC, EU Statistics on Income and Living Conditions; FI, Finland; FR, France; HU, Hungary; IE, Ireland; IS, Iceland; IT, Italy; LT, Lithuania; LU, Luxembourg; LV, Latvia; NL, the Netherlands; NO, Norway; PL, Poland; PT, Portugal; SE, Sweden; SK, Slovakia.

intermediately educated, 75% were employed and 27% had a chronic illness. The Eastern region had a large intermediate educated group (64%), whereas the Southern and Anglo-Saxon regions had a relatively large low-educated group (28% and 38%, respectively) (table 1). The Southern region had the lowest labour force participation, and the lowest percentage of people with a chronic illness. The Nordic region had the highest employment but also the highest prevalence of people with a chronic illness.

Labour force participation over the period 2005–2014 among men without a chronic illness remained stable for intermediate and high-educated persons but decreased by about 5% among those with lower education (figure 1A). Within each educational group, labour force participation among men with a chronic illness was substantially lower than among men without illness, with the largest difference among the lowest educated persons. For women, the labour force participation increased slightly for most groups over the period 2005–2014 (figure 1B). Women had a consistently lower labour force participation than men with increasing differences by lower educational level. Women with a chronic disease had substantially lower labour force participation than women without a chronic illness.

The absolute differences in labour force participation between persons with and without a chronic illness increased with lower educational level among men and women (table 2). The absolute differences were larger for men than for women. For low-educated persons, the absolute differences in labour force participation between men with and without a chronic illness were higher in the Nordic, Anglo-Saxon and Eastern region (32.9%–35.3%) than in the Continental and Southern region (21.5%–26.1%). Among women a comparable pattern was observed.

A similar pattern was found for relative differences in health-related labour force participation (table 3). Overall, low-educated men with a chronic illness were 1.4 to 1.9 times more likely to be outside of paid employment than low-educated men without a chronic illness, whereas this was 1.1 to 1.2 times within the high educational group. Relative differences in labour force participation among low-educated men without and with a chronic illness were the highest in the Anglo-Saxon and Eastern region (1.79–1.94). For men with high education, modest regional differences were found in labour force participation between those without and with a chronic illness. Comparable patterns were observed for women.

Trends in absolute differences in labour force participation showed stable inequalities for men in all educational groups in most regions (table 4, see online supplementary tables 1 and 2). Among men, significant narrowing of absolute health-related inequalities in paid employment was observed for low and high-educated persons in the Southern region over the period 2005–2014. For women, significant widening of absolute health-related inequalities in paid employment were observed for low and intermediate-educated women in the Continental and Anglo-Saxon region. In the Eastern region, significant

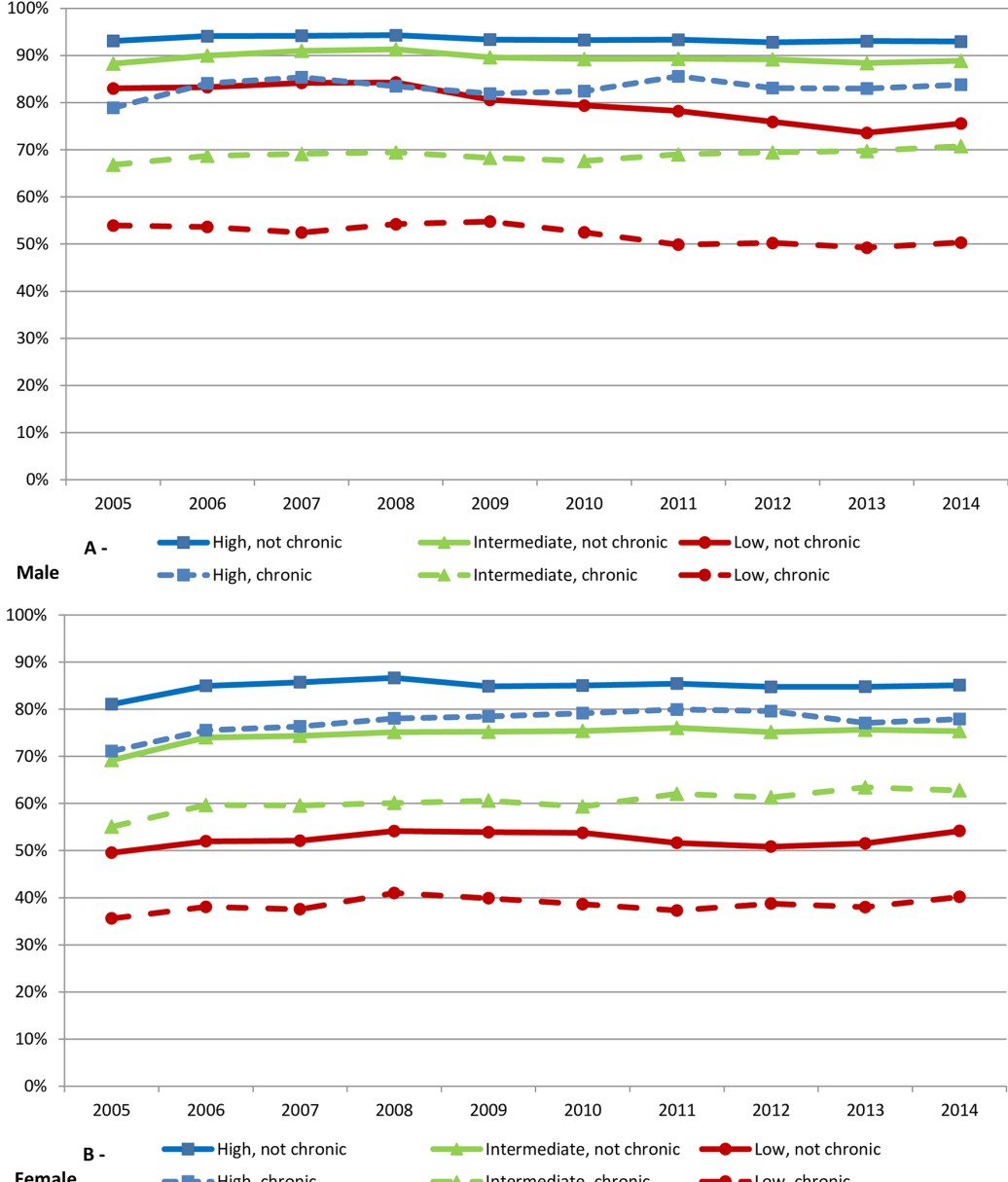

**Figure 1** Proportion in paid employment with and without chronic illness by educational level—men (above) and women (below) in 26 European countries (2005–2014).

widening in inequalities was observed for low-educated women, while narrowing inequalities were observed for high-educated women.

Trends in relative differences in labour force participation were constant among high-educated men and widened in low-educated men (table 5, see online supplementary tables 3 and 4). Among men, significant widening of relative differences were observed for lower educated persons in the Nordic and Continental region of respectively 1% and 2.6% over the period 2005–2014. Significant narrowing of relative health-related inequalities in paid employment was observed among higher educated men in the Continental and Southern region (−0.3% and −0.6%). Trends for women were similar. In most regions, relative health-related inequalities in paid employment were widening for low-educated women (between 1.0%

and 3.7%), although in the Southern region, there was a small narrowing (−0.7%). For high-educated women there was no clear trend.

Figure 2 presents the inequalities between low and high-educated persons for the influence of a chronic illness on being in paid employment within each educational group. The excess relative difference among low-educated persons across 26 European countries varied between 50% and 100% among men and between 58% and 100% among women when compared with high-educated persons. The variation in educational inequalities in labour market position of those with a chronic disease was larger within European regions than between European regions, indicating substantial differences between countries with supposedly similar welfare regimes.

**Table 2** Absolute differences in paid employment between persons with and without a chronic illness by educational level in European regions in 2014

| Region in Europe | Absolute difference (%) (95% CI) | | |
| --- | --- | --- | --- |
| | Within low education (%) | Within intermediate education (%) | Within high education (%) |
| **Male** | | | |
| Nordic | 32.9 (28.9 to 36.8) | 18.6 (16.4 to 20.9) | 11.1 (8.5 to 13.7) |
| Continental | 26.1 (23.6 to 28.5) | 17.2 (15.7 to 18.6) | 8.2 (6.4 to 10.0) |
| Anglo-Saxon | 35.3 (31.8 to 38.7) | 17.1 (13.1 to 21.1) | 13.4 (9.9 to 17.0) |
| Southern | 21.5 (19.7 to 23.3) | 11.2 (8.9 to 13.5) | 5.0 (2.2 to 7.9) |
| Eastern | 33.5 (30.8 to 36.3) | 25.5 (24.2 to 26.7) | 9.8 (7.2 to 12.4) |
| **Female** | | | |
| Nordic | 31.0 (26.2 to 35.7) | 20.1 (17.3 to 22.9) | 9.6 (7.1 to 12.0) |
| Continental | 20.4 (17.7 to 23.0) | 15.2 (13.5 to 16.9) | 9.4 (7.3 to 11.6) |
| Anglo-Saxon | 29.8 (25.8 to 33.8) | 22.5 (18.5 to 26.6) | 15.8 (12.2 to 19.4) |
| Southern | 10.0 (8.0 to 11.9) | 6.5 (4.0 to 8.9) | 5.8 (2.9 to 8.6) |
| Eastern | 23.7 (20.7 to 26.6) | 20.6 (19.3 to 22.0) | 6.4 (4.2 to 8.6) |

## DISCUSSION

Within Europe, persons with a chronic disease were less often in paid employment than persons without a chronic disease in the period 2005–2014. The low education group had the largest absolute and relative differences in paid employment between persons with and without a chronic illness. For men these differences were more pronounced than for women. The Nordic, Anglo-Saxon and Eastern regions had substantially higher relative and absolute differences in paid employment than in the Continental and Southern region. The relative position on the labour market for persons with a chronic illness worsened in the past 10 years among low-educated persons and remained fairly stable for high-educated persons.

The risk of being outside of paid employment with a chronic illness is higher for persons with a low education than those with a high education. Previous studies have shown comparable results in specific countries[9 11 16 18] or across Europe in one particular year.[27 28] Our study expands this knowledge by demonstrating that these patterns are consistent over time and European countries and by illustrating that health-related educational inequalities in paid employment have widened rather than narrowed in the past decade.

**Table 3** Relative differences in paid employment between persons with and without a chronic illness by educational level in European regions in 2014

| Region in Europe | Relative difference (PR) (95% CI) | | |
| --- | --- | --- | --- |
| | Within low education | Within intermediate education | Within high education |
| **Male** | | | |
| Nordic | 1.61 (1.46 to 1.77) | 1.25 (1.21 to 1.30) | 1.13 (1.10 to 1.17) |
| Continental | 1.49 (1.40 to 1.58) | 1.24 (1.21 to 1.27) | 1.09 (1.07 to 1.12) |
| Anglo-Saxon | 1.79 (1.64 to 1.95) | 1.25 (1.17 to 1.33) | 1.17 (1.12 to 1.23) |
| Southern | 1.43 (1.38 to 1.49) | 1.16 (1.12 to 1.20) | 1.06 (1.03 to 1.09) |
| Eastern | 1.94 (1.78 to 2.11) | 1.42 (1.39 to 1.45) | 1.11 (1.08 to 1.15) |
| **Female** | | | |
| Nordic | 1.65 (1.48 to 1.84) | 1.30 (1.24 to 1.36) | 1.12 (1.09 to 1.15) |
| Continental | 1.48 (1.39 to 1.58) | 1.24 (1.20 to 1.27) | 1.12 (1.09 to 1.15) |
| Anglo-Saxon | 1.89 (1.70 to 2.11) | 1.44 (1.33 to 1.57) | 1.23 (1.17 to 1.30) |
| Southern | 1.28 (1.21 to 1.34) | 1.11 (1.07 to 1.16) | 1.08 (1.04 to 1.12) |
| Eastern | 1.76 (1.62 to 1.92) | 1.37 (1.33 to 1.40) | 1.08 (1.05 to 1.10) |

PR, prevalence ratio.

**Table 4** Trends in absolute health-related inequalities in paid employment in European regions over the period 2005–2014

| Region in Europe | Trend* | | |
| --- | --- | --- | --- |
| | Low education | Intermediate education | High education |
| Male | | | |
| Nordic | **0.009 (0.004 to 0.014)** | 0.001 (−0.002 to 0.003) | 0.001 (−0.003 to 0.005) |
| Continental | 0.003 (−0.001 to 0.007) | 0.001 (0.000 to 0.003) | **−0.002 (−0.005 to 0.000)** |
| Anglo-Saxon | −0.003 (−0.013 to 0.007) | 0.005 (−0.002 to 0.012) | 0.003 (−0.002 to 0.007) |
| Southern | **−0.005 (−0.009 to -0.002)** | −0.003 (−0.006 to 0.001) | **−0.005 (−0.009 to -0.001)** |
| Eastern | 0.002 (−0.004 to 0.009) | −0.001 (−0.004 to 0.002) | 0.000 (−0.003 to 0.003) |
| Female | | | |
| Nordic | 0.006 (−0.003 to 0.015) | −0.001 (−0.005 to 0.003) | −0.002 (−0.007 to 0.002) |
| Continental | **0.008 (0.004 to 0.011)** | **0.004 (0.003 to 0.006)** | 0.002 (−0.002 to 0.005) |
| Anglo-Saxon | 0.006 (0.000 to 0.013) | **0.008 (0.002 to 0.014)** | 0.002 (−0.005 to 0.008) |
| Southern | −0.001 (−0.004 to 0.002) | 0.001 (−0.003 to 0.005) | −0.003 (−0.008 to 0.003) |
| Eastern | **0.004 (0.000 to 0.008)** | 0.002 (−0.001 to 0.004) | **−0.003 (−0.005 to 0.000)** |

*Trend describes the widening (positive value) or narrowing (negative value) of the absolute difference of proportion in paid employment between participants with and without a chronic illness.
The bolded values are the values that are significant (p <0.05).

Men and women with a chronic illness had less access to the labour market than their healthy colleagues, especially among lower educated persons. Possible explanations for the observed educational inequality relate to severity of chronic illnesses and quality of jobs. In this study, chronic illness could not be differentiated by diagnosis and severity. It may be hypothesised that lower educated workers more often suffer from chronic diseases with functional limitations than high-educated workers.[29 30] Having a chronic illness and functional limitations follows a social gradient in our data (high educated 49%, low educated 73%). The education inequality may also be explained by differences

in strenuous work and other job characteristics that may interfere with health problems.[31 32]

A few studies have demonstrated that strenuous work conditions hamper workers with a chronic illness to remain in paid employment. Among Finnish workers, physical work load strongly mediated the association between musculoskeletal disorders and exit from paid employment.[33] A Dutch study attributed 12% of educational differences in disability benefits to work characteristics.[34] Another Dutch study showed that high autonomy, high social support and low psychosocial job demands strongly moderated the association between

**Table 5** Trends in relative health-related inequalities in paid employment in European regions over the period 2005–2014

| Region in Europe | Trend* | | |
| --- | --- | --- | --- |
| | Low education | Intermediate education | High education |
| Male | | | |
| Nordic | **0.026 (0.012 to 0.040)** | 0.002 (−0.003 to 0.007) | 0.001 (−0.004 to 0.006) |
| Continental | **0.010 (0.001 to 0.020)** | 0.002 (−0.001 to 0.005) | **−0.003 (−0.007 to 0.000)** |
| Anglo-Saxon | 0.014 (−0.024 to 0.053) | 0.014 (−0.001 to 0.029) | 0.005 (−0.002 to 0.012) |
| Southern | −0.001 (−0.007 to 0.004) | −0.002 (−0.007 to 0.002) | **−0.006 (−0.011 to -0.001)** |
| Eastern | 0.013 (−0.008 to 0.035) | −0.004 (−0.010 to 0.002) | 0.000 (−0.004 to 0.004) |
| Female | | | |
| Nordic | 0.028 (−0.004 to 0.060) | −0.001 (−0.008 to 0.006) | −0.003 (−0.009 to 0.002) |
| Continental | **0.015 (0.002 to 0.028)** | **0.004 (0.001 to 0.007)** | 0.001 (−0.004 to 0.007) |
| Anglo-Saxon | 0.037 (−0.003 to 0.077) | **0.020 (0.006 to 0.034)** | 0.003 (−0.007 to 0.013) |
| Southern | −0.007 (−0.016 to 0.001) | 0.003 (−0.004 to 0.011) | −0.003 (−0.010 to 0.005) |
| Eastern | 0.010 (−0.011 to 0.031) | 0.000 (−0.007 to 0.006) | −0.002 (−0.005 to 0.002) |

*Trend describes the widening (positive value) or narrowing (negative value) of the relative difference of prevalence ratios in paid employment between participants with and without a chronic illness.
The bolded values are the values that are significant (p <0.05).

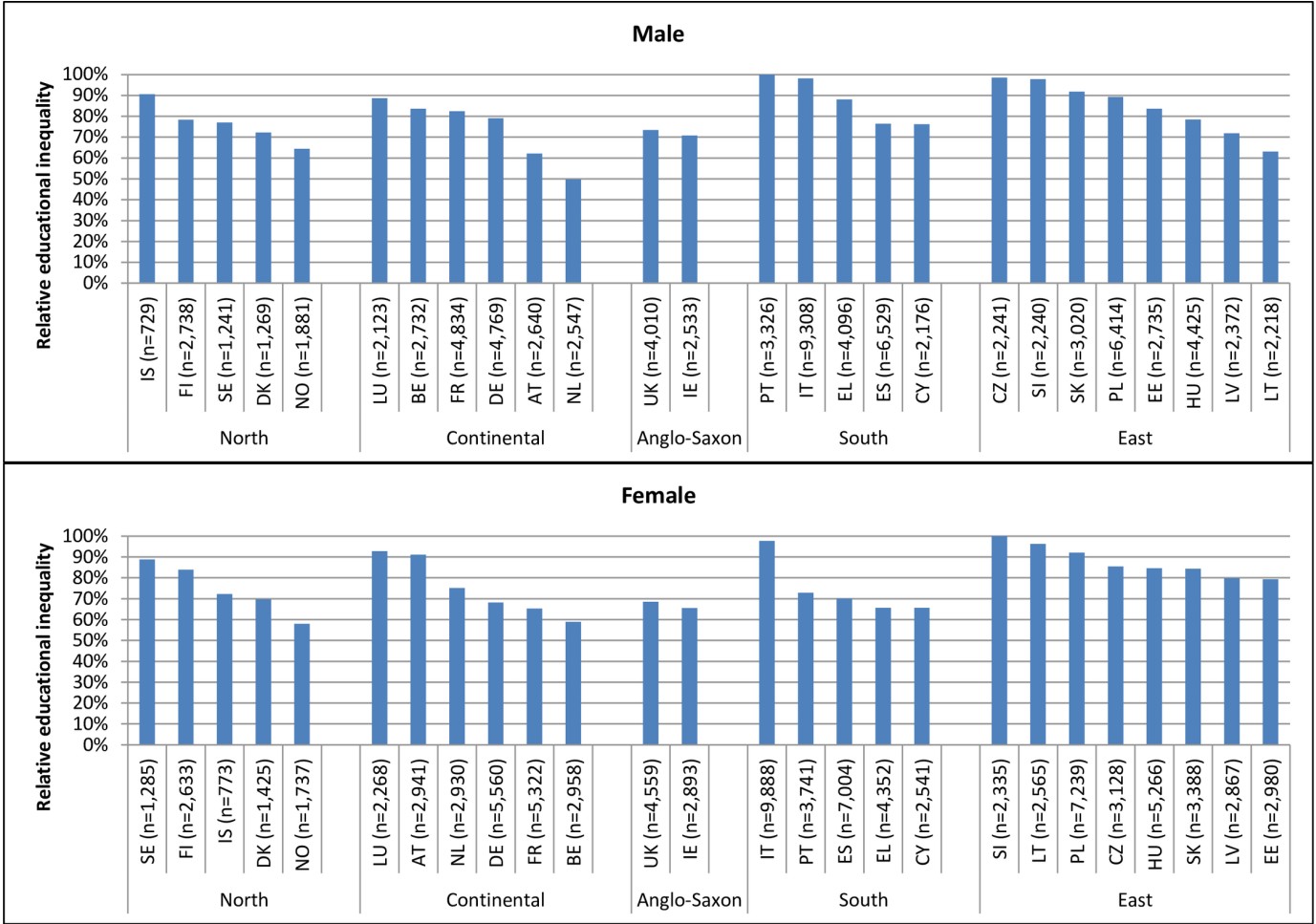

**Figure 2** Relative inequality in paid employment among persons with a chronic illness comparing low-educated persons with high-educated persons.

chronic health conditions and exit through disability benefits.[35] Lower socioeconomic jobs more often have unfavourable work characteristics.[36] Therefore, the ability to remain in paid employment is reduced for low-educated persons with a chronic illness.

Health-related absolute and relative differences in paid employment across educational groups were consistently lower in the Southern region than in other European regions. A few studies[12 18 27] have noted larger absolute and relative differences in labour force participation between persons with and without a chronic illness in regions with low employment protection compared with regions with high employment protection. Our results in the Southern region, which is characterised by high employment protection, corroborates these previous studies. However, the educational inequalities in paid employment depicted in figure 2 showed a larger variation between countries in the same region than between different regions. This suggests that other social, legal and economic conditions, such as availability of social benefits, also play a role in being in paid employment of people with a chronic illness. More studies are necessary to evaluate the influence of various national indicators of social, cultural and economic conditions on the variation in health-related educational inequalities in paid employment.

The absolute and relative disadvantages in labour force participation among those with a chronic disease were more pronounced for men than for women in most welfare states. In the Southern welfare state regime, the absolute and relative differences were smaller for women, but in the Nordic welfare state regime, the two groups showed similar patterns. These regional differences may partly be explained by institutional support, that is, the full-employment policies of the Nordic region, facilitating the integration of women into the labour force by providing for example child care, although often the decision to be active or not is strongly affected by education and fertility choices.[37 38] The high labour force participation of women in Nordic welfare states can contribute to higher relative differences between those with and without a chronic illness compared with the Southern region. This is in line with previous studies stating that in terms of socioeconomic inequalities, the relative inequalities do not appear to be smallest in more redistributive countries such as the Scandinavian welfare state, but that the absolute health of all social classes is often better.[39]

The trends observed in this study showed that over the period 2005 to 2014, educational inequalities in paid employment of persons with a chronic illness have widened among low-educated persons. In the recent decades, precarious employment, such as temporary contracts and part-time work, have become more common in Europe.[40 41] Further research is necessary to evaluate whether the increase in precarious employment could be an explanation for increasing educational inequalities in paid employment.

The trend analysis might be influenced by the differential effect of the economic crisis across regions. The Continental and Eastern region have had an increase in persons in paid employment over the period 2005–2014, while the Southern region had a decrease of persons in paid employment due to the economic crisis. In periods of high and rapid increase in unemployment rates, workers in good health will also lose their jobs and, thus, the influence of ill health on paid employment will attenuate.[42] This mechanism may explain why we found a decreasing trend in the Southern region. Furthermore, health may have worsened during the economic crisis. For persons with existing health issues, this could have led to more difficulties maintaining paid employment. A recent study found an increase in absolute inequalities in health between educational groups after the recent economic crisis in the Continental and Anglo-Saxon region.[43] This may explain the increasing trends we found in these regions.

### Strengths and limitations

Data were derived from the EU-SILC. A strength of the current study is the use of repeated cross-sectional data from a large number of European countries for which the outcomes are harmonised. By using Cox regression instead of logistic regression, differences in labour force attachment in EU-SILC across countries will not affect the results of the statistical analysis. The repeated cross-sectional surveys present a snapshot of the labour force situation of each region for the respective years, but causal relations cannot be investigated in depth, as other studies with cohort information have done.[44 45] By using cross-sectional data in this study, the effects of having a chronic illness on paid employment (selection) and unemployment causing ill health, such as chronic illness (causation), cannot be distinguished.

A limitation of using EU-SILC data is the variation in mode of data collection, translations and cultural interpretation.[27] Further limitations are the use of self-reported health and labour market measures which could vary by country, socioeconomic position and cultural beliefs.[13] However, a number of studies have shown that self-rated measurements of health are strongly predictive for objective health measures, such as mortality.[46 47] Comparisons in self-rated health measures between different cultures do need to be made with caution.[48]

Sixteen per cent of our sample did not respond to the question on chronic illness. These non-respondents had a similar age, were higher educated and more often employed than respondents. Based on their characteristics, non-respondents were less likely to have a chronic illness. Although it is difficult to predict the influence of this selective reporting, the slight under-representation of subjects with a chronic disease may have resulted in an underestimation of the association between having a chronic illness and being in paid employment.

### Policy implications

Men and women with a chronic illness have considerable less access to the labour market than their healthy colleagues, especially among lower educated persons. Their position in the labour market seems to have worsened rather than improved in the past decade. Persons with a chronic illness without a paid job are at risk for further deterioration of their health and economic marginalisation. More effective policy measures are needed to reduce the gap in paid employment between those with and without a chronic illness, especially in the low education group, in all European regions.

**Correction notice** This article has been corrected since it first published online. The open access licence type has been amended.

**Acknowledgements** We kindly acknowledge the help of the Microdata Access team from Eurostat for provision of the EU-SILC datasets to our research entity (2014/044/NL). We kindly thank the WORKLONG Impact Group, in particular Lars Andersen, professor at the Danish National Research Centre for the Working Environment and Manuel Flores, economist at the Directorate for Employment, Labour and Social Affairs at the Organisation for Economic Co-operation and Development, for their comments on the draft of the paper. We also thank the organisation of the European Public Health Conference 2018, where the results of this study have been presented. The submitted abstract has been published in the European Journal of Public Health, https://doi.org/10.1093/eurpub/cky213.284.

**Contributors** JLDS prepared the data, conducted the analysis and drafted and revised the paper. MS and KMOH participated in the analysis and commented on the paper. AB conceptualised the study, participated in the analysis and commented on the paper and is the guarantor. All authors approved the final version.

**Funding** This work was conceived with financial support from award no. 208060001 by ZonMW within the Joint Programming Initiative More Years Better Lives (WORKLONG project) framework. Additional financial support was received through EIT Health.

**Competing interests** None declared.

**Patient consent for publication** Not required.

**Provenance and peer review** Not commissioned; externally peer reviewed.

**Data sharing statement** Anonymised, non-identifiable participant-level survey data are freely available for academic researchers after successful application from Eurostat.

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
