## [Reviewer comments · BMJ Open]

ARTICLE DETAILS

TITLE (PROVISIONAL)	Health-related educational inequalities in paid employment across 26 European countries in 2005-2014: a repeated cross-sectional study
AUTHORS	Schram, Jolinda; Schuring, Merel; Oude Hengel, Karen; Burdorf, Alex

VERSION 1 - REVIEW

REVIEWER	Teresa Leão National School of Public Health, Portugal
REVIEW RETURNED	13-Jul-2018

GENERAL COMMENTS	Notes to the authors: 1. The rationale/focus of the paper should be more clearly stated in the introduction. The paper seems to assess (1) if there are inequalities of access to paid employments by health status, and (2) if these inequalities are socially patterned. This second point is mentioned in the lines 14-15 and 33 of page 4, but the sentence in line 14-15 seems inserted in the middle of the first point of the rationale. Also, it is not clear if authors' concerns related to the consequences of low labour force participation by people with worst health status and by low educated groups are economic or health concerns, or both. If the Introduction seems to focus mostly on the first consequence ("the challenge is ensuring that those of working age are actually participating in the labour force"), the Policy Implications seems to focus on the second ("persons with a chronic illness without a paid job are at risk for further deterioration of their health and economic marginalisation"). As both consequences have relevant policy implications, both should be more clearly stressed in the Introduction and Policy Implications' sections. 2. The outcome should be better defined. First, several expressions are used through the text as synonyms (employment/unemployment, paid/non-paid work, part/not part of the labour market,...), but they enclose different concepts, and reflect different concerns. If the focus is purely the economic consequences (fewer contributions because these groups are excluded), the authors may focus on non-paid work as this will not provide contributions for the state (unemployment, work with no salary, students, retirement, etc.). If the focus is on the health consequences (people are negatively selected because they are ill, which puts them in further risk of being ill), which seems to be the case, the authors should probably focus on unemployment. Second, the authors include in the "non-paid" group unemployed, inactive, retired, disabled, students, military groups, etc. Though,
---

people who retire before 60 years old usually do so because of illness; they are not part of the “paid employment” group because they are ill, not because of the employer has discarded them or they were not selected for a work. In this case, they probably have a retirement benefit, which protect their health and economic wellbeing. Similarly, being a student is a decision that is not directly linked to rejection from the labour market, and it is independent of illness. Similarly, military service, even when compulsory, is not due to labour market rejection, is not linked with illness, and this group frequently receive a salary, even if low. Depending on the rationale of the paper, the authors should better explain how the outcome is defined and why, and/or present literature supporting their reasons.

3. The expression “health-related educational inequalities” can be confusing. This paper is about inequalities in access to work because of health status, by education level; not inequalities in access to education because of health status. I would avoid to use it as much as possible, either using through the text the expression “health-related educational inequalities in paid employment”, or rewording it in a more transparent way (for example using “trends in inequalities in access to paid employment, by health and education status”).

4. I would like to see the rates of labour participation, or the absolute and relative differences, by year and by region, in a table for supplementary material. Figure 1 suggests some different sizes and variations in the rates of labour participation in different time periods, pre, during, and after the economic crisis. Though its duration and beginning/ending was not similar in all countries, it generally followed a welfare-pattern. Presenting your results in a more disaggregated way can probably help explaining the trends in the different regions.

5. Another explanation for the trends of absolute and relative inequalities may complement the one presented by the authors in page 12:

Health may have worsened during the economic crisis period. The worsening in the health status of persons that already had the burden of chronic diseases, may have led to a higher difficulty to stay in paid jobs, raising health-related inequalities, especially among low SES groups. Some welfare states experienced more severe worsening in health among low SES than others: in a previous study, we found significant trend in widening education-related health inequalities in the Bismarkian and Anglosaxon countries for absolute inequalities, and in the Post-Communist countries and Bismarkian, for relative ones. I think these results help discuss these papers’ trends (see Leão T, Campos-Matos I, Bamba C, Russo G, Perelman J. Welfare states, the Great Recession and health: Trends in educational inequalities in self-reported health in 26 European countries. PLoS One. 2018;13(2):1–14.).

6. The authors hypothesised in the Discussion that “lower educated workers more often suffer from chronic diseases with functional limitations than high educated workers“, but functional limitations were not included as a health variable. Why did the authors not use it, as it is measured by the EU-SILC? I would suggest adjusting the analyses for functional limitations, in order to

	try to explain better the differences in access to paid employment by people with chronic disease. 7. Also, it is not clear why the authors grouped age in 10-year categories, instead of using it as continuous variable. It is also not mentioned if the analyses were adjusted for country fixed effects. 8. More recent studies have divided the Eastern countries into two groups: Post-Communists and Former-USSR. These differ on the “levels of economic growth, inflation, social well-being and egalitarianism” (Campos-Matos I, Kawachi I. Social mobility and health in European countries: Does welfare regime type matter? Soc Sci Med. 2015 Oct;142:241–8). These characteristics may influence the outcomes of this study. 9. Please explain the rationale behind the choice of dividing the job access-inequalities by education instead of by occupational class (as did by Minton et al, 2012). 10. The authors should pay special attention to reverse causality, especially when considering the path between health and unemployment. This issue should be stated in the Introduction, as well as the reasons that support the authors on pursuing this study despite that issue.
--	--

REVIEWER	Punzo, Gennaro University, Italy
REVIEW RETURNED	25-Jul-2018

GENERAL COMMENTS	Summary Purpose of this paper: to study trends in health-related inequalities in labour force participation and differences between men and women by education level and across European countries using cross-sectional data (EU-SILC, 2005-2014). The authors consider information on work status, chronic illness, educational background, age, and gender, and perform Cox regression to sketch differences among five different groups of European States. Comments 1) In my interpretation, what this paper actually does is to estimate the Cox proportional hazard model to calculate relative differences in labour force participation among workers with and without a chronic disease across educational groups, but just loosely attempting to relate the findings to institutional aspects of European countries. In this perspective, the paper should take a deeper look at the role of institutional settings and social benefits in understanding the dynamics of health-related educational inequalities in paid employment. In other words, the Authors should pay more attention to discuss their results in light of the main characteristics of national education and social security systems and, finally, welfare state regimes. 2) Indeed, with reference to the previous comment (1), the comparison between 2005 and 2014 does not take into account that the dynamics of changes observed may be influenced by economic and political factors. In particular, it is not considered that changes in the political, economic and institutional conditions of each country or groups of countries might have affected the differences over the years. Moreover, the role of different public policies and institutional aspects that can be responsible for these changes is not discussed enough.
--

	3) In order to better frame the topic, a link with the existing literature and the contribution of the paper should be clarified. References from the scientific literature should be strengthened. 4) The modelling strategy is not clearly motivated and described. The Authors should better explain how the Cox regression overcomes the main limits of the traditional logistic regression in their specific field of study. In other words, I think the advantages that could derive by the use of Cox regression instead of the classical logit/probit models are not too much emphasised. 5) Moreover, some additional lines should be devoted to explain how potential problems of sample selection regarding the dichotomy being/not being in the labour force and the way through which it could be treated. I should stress a little bit more the crucial point of the proposal that is the use of Cox regression. 6) In the discussions, some interesting food for thought for future research is provided. The results, though, imply policy implications of which no suggestions are given or they are very poor.
--	--

VERSION 1 – AUTHOR RESPONSE

Reviewer(s)' Comments to Author:

Reviewer: 1

Reviewer Name: Teresa Leão

Institution and Country: National School of Public Health, Portugal

Please state any competing interests or state 'None declared': None declared

Please leave your comments for the authors below

Notes to the authors:

Comment #1: The rationale/focus of the paper should be more clearly stated in the introduction. The paper seems to assess (1) if there are inequalities of access to paid employments by health status, and (2) if these inequalities are socially patterned. This second point is mentioned in the lines 14-15 and 33 of page 4, but the sentence in line 14-15 seems inserted in the middle of the first point of the rationale.

Also, it is not clear if authors' concerns related to the consequences of low labour force participation by people with worst health status and by low educated groups are economic or health concerns, or both. If the Introduction seems to focus mostly on the first consequence ("the challenge is ensuring that those of working age are actually participating in the labour force"), the Policy Implications seems to focus on the second ("persons with a chronic illness without a paid job are at risk for further deterioration of their health and economic marginalisation"). As both consequences have relevant policy implications, both should be more clearly stressed in the Introduction and Policy Implications' sections.

Reply to comment #1: Order of the sentence line 14-15 has been changed in the second paragraph of the introduction, the sentence [Educational inequalities in morbidity...across European countries] has been moved from the beginning of the paragraph to in between, making the order more logical.

The focus is on whether people with health issues are in paid employment, whether it is socially patterned and whether there have been changes in the period 2005-2014. We know, based on previous research, that not being in paid employment can lead to further isolation and often

deteriorating of health, even of people who are already ill. While this is not the focus of the current paper, since we cannot evaluate this in our cross-sectional dataset, the argument is important as contextual information and it has been introduced in the introduction. Added line: “Leaving paid employment may further impact health negatively”, in the first paragraph. The sentence in the policy implications referring to “despite policies and programmes to improve” has been removed, also from the abstract. The first sentence from the first paragraph in the introduction, [“Many countries ... in paid employment”] has also been removed.

Comment #2: The outcome should be better defined. First, several expressions are used through the text as synonyms (employment/unemployment, paid/non-paid work, part/not part of the labour market,...), but they enclose different concepts, and reflect different concerns. If the focus is purely the economic consequences (fewer contributions because these groups are excluded), the authors may focus on non-paid work as this will not provide contributions for the state (unemployment, work with no salary, students, retirement, etc.). If the focus is on the health consequences (people are negatively selected because they are ill, which puts them in further risk of being ill), which seems to be the case, the authors should probably focus on unemployment.

Second, the authors include in the “non-paid” group unemployed, inactive, retired, disabled, students, military groups, etc. Though, people who retire before 60 years old usually do so because of illness; they are not part of the “paid employment” group because they are ill, not because of the employer has discarded them or they were not selected for work. In this case, they probably have a retirement benefit, which protect their health and economic wellbeing. Similarly, being a student is a decision that is not directly linked to rejection from the labour market, and it is independent of illness. Similarly, military service, even when compulsory, is not due to labour market rejection, is not linked with illness, and this group frequently receive a salary, even if low.

Depending on the rationale of the paper, the authors should better explain how the outcome is defined and why, and/or present literature supporting their reasons.

Reply to comment #2:

As mentioned in our previous comment, the focus is on whether people with health problems are in paid employment or outside of paid employment, whether this is socially patterned and whether there have been changes in the period 2005-2014. Therefore, we did not focus only on disability or unemployment. We decided to provide a broad overview regardless of the underlying reasons for not being in paid employment. We know that being in poor health is related to unemployment and disability, but that it can also affect the decision to retire early. Since different exit routes will influence each other differently across countries, throughout the paper we have made the distinction between being in paid employment and being outside of paid employment. Also, we have changed the heading of the variable in the method section from “Labour force participation” to “Employment status”.

Comment #3: The expression “health-related educational inequalities” can be confusing. This paper is about inequalities in access to work because of health status, by education level; not inequalities in access to education because of health status. I would avoid to use it as much as possible, either using through the text the expression “health-related educational inequalities in paid employment”, or rewording it in a more transparent way (for example using “trends in inequalities in access to paid employment, by health and education status”).

Reply to comment #3: Thank you for your suggestion, we have adjusted “health-related educational inequalities” to “health-related educational inequalities in paid employment”, see track changes in paper.

Comment #4: I would like to see the rates of labour participation, or the absolute and relative differences, by year and by region, in a table for supplementary material. Figure 1 suggests some

different sizes and variations in the rates of labour participation in different time periods, pre, during, and after the economic crisis. Though its duration and beginning/ending was not similar in all countries, it generally followed a welfare-pattern. Presenting your results in a more disaggregated way can probably help explaining the trends in the different regions.

Reply to comment #4: Thank you for your suggestion We have added the absolute and relative differences in four supplementary tables by gender, by year and by region (ref. added in results section).

Comment #5: Another explanation for the trends of absolute and relative inequalities may complement the one presented by the authors in page 12:

Health may have worsened during the economic crisis period. The worsening in the health status of persons that already had the burden of chronic diseases, may have led to a higher difficulty to stay in paid jobs, raising health-related inequalities, especially among low SES groups. Some welfare states experienced more severe worsening in health among low SES than others: in a previous study, we found significant trend in widening education-related health inequalities in the Bismarkian and Anglosaxon countries for absolute inequalities, and in the Post-Communist countries and Bismarkian, for relative ones. I think these results help discuss these papers' trends (see Leão T, Campos-Matos I, Bamba C, Russo G, Perelman J. Welfare states, the Great Recession and health: Trends in educational inequalities in self-reported health in 26 European countries. PLoS One. 2018;13(2):1–14.).

Reply to comment #5: Thank you for pointing us towards the paper. We have added the proposed explanation for worsening of health to the discussion section: "Furthermore, health may have worsened during the economic crisis. For persons with existing health issues, this could have led to more difficulties maintaining paid employment. A recent study found an increase in absolute inequalities in health between educational groups after the recent economic crisis in the Continental and Anglo-Saxon region (41). This may explain the increasing trends we found in these regions. "

Comment #6: The authors hypothesised in the Discussion that "lower educated workers more often suffer from chronic diseases with functional limitations than high educated workers", but functional limitations were not included as a health variable. Why did the authors not use it, as it is measured by the EU-SILC? I would suggest adjusting the analyses for functional limitations, in order to try to explain better the differences in access to paid employment by people with chronic disease.

Reply to comment #6: Using the variable functional limitations would be of interest, but this is outside of the scope of the study as the material presented is already complex and diverse. This study is aimed at providing the broader picture of chronic diseases and is not about comparing different definitions of ill health.

Comment #7: Also, it is not clear why the authors grouped age in 10-year categories, instead of using it as continuous variable. It is also not mentioned if the analyses were adjusted for country fixed effects.

Reply to comment #7: We did not want to assume a particular pattern (such as a linear association), of the association of age with chronic disease and being out of paid employment, and, thus, preferred to include age as categorical variable in the analysis. This assumption was compared with the suggested assumption of a linear association and we found almost exactly similar results. The analyses on European regions were not adjusted for country fixed effects, but the analysis for Figure 2 was stratified by country.

Comment #8: More recent studies have divided the Eastern countries into two groups: Post-Communists and Former-USSR. These differ on the "levels of economic growth, inflation, social well-being and egalitarianism" (Campos-Matos I, Kawachi I. Social mobility and health in European

countries: Does welfare regime type matter? Soc Sci Med. 2015 Oct;142:241–8). These characteristics may influence the outcomes of this study.

Reply to comment #8: Thank you for your suggestion,. We know that different authors have suggested different classification, and the group of Eastern European countries could indeed be divided into former USSR states, in our case, the Baltics and the other East European states. In our study we follow an earlier distinction by Bambra & Eikemo (2009), in which welfare state regimes are based on the amount of social protection, such as employment protection and income security. The paper you refer to, in turn refers to the paper by Fenger (2007) “Welfare regimes in central and Eastern Europe: incorporating post-communist countries in a welfare regime typology”, where these two Eastern European welfare regimes originate from cluster analysis. The 19 variables that were included in the cluster analysis hardly included social security (spending on social protection, revenue from social contributions) and therefore the 2 two group distinction seems not compatible with the distinction of welfare state regimes we have used in our paper.

Comment #9: Please explain the rationale behind the choice of dividing the job access-inequalities by education instead of by occupational class (as did by Minton et al, 2012).

Reply to comment #9: Inequalities in labour force participation and health are always defined along the line of a socio-economic group marker, either occupational group, educational group or income quintile. Since our outcome is being in paid employment or outside, there may not have an occupational class available for those who have never been in paid employment or have been outside of paid employment for a long time. Furthermore, education is the key socio-economic variable that is well defined and standardized in the EU SILC data. The variable is easily comparable across countries, and also with the large body of studies on health inequalities, whereby education is the most used classification in European studies.

Comment #10: The authors should pay special attention to reverse causality, especially when considering the path between health and unemployment. This issue should be stated in the Introduction, as well as the reasons that support the authors on pursuing this study despite that issue.

Reply to comment #10: The focus on the paper is not the bidirectional relationship between changes in health and transitions between paid employment and unemployment, but health and being in paid employment. We have mentioned mechanisms of selection and causation in the introduction, although not the terminology as such, since the current study (repeated cross-sectional surveys) does not focus on causal mechanisms, but inequalities and trends in inequalities. The latter information is almost completely limited to a few studies on single countries. Thus, we think that our current analysis adds to the insight into health-related educational inequalities in paid employment across Europe. Further work on selection and causation mechanisms should use longitudinal surveys.

Reviewer: 2

Reviewer Name: Gennaro Punzo

Institution and Country: Univ Naples Parthenope, Italy

Comments:

Comment # 1:

In my interpretation, what this paper actually does is to estimate the Cox proportional hazard model to calculate relative differences in labour force participation among workers with and without a chronic disease across educational groups, but just loosely attempting to relate the findings to institutional

aspects of European countries. In this perspective, the paper should take a deeper look at the role of institutional settings and social benefits in understanding the dynamics of health-related educational inequalities in paid employment.

In other words, the Authors should pay more attention to discuss their results in light of the main characteristics of national education and social security systems and, finally, welfare state regimes.

Reply to comment #1:

The aim of the paper is to determine whether people with health issues are in paid employment, whether the pattern of being in paid employment differs in educational groups and whether there have been changes in these patterns in the period 2005-2014. Studies investigating these patterns over time are almost exclusively on single countries. We investigate the dynamics of these inequalities over time, without linking them to specific country characteristics but distinguishing between different European regions which influences our findings.

We have rewritten some parts of the introduction to stress more the differences between regions and the patterns investigated in the current study, rather than the underlying mechanisms of the welfare state.

New paragraph:

The differential effect of socioeconomic position on the association between chronic illness and paid employment may be modified by national social and economic conditions, such as labour market policies (14). European countries can be categorized in five regions: the Nordic, Continental, Anglo-Saxon, Southern, and Eastern regions (15, 16). This typology reflects general differences in employment protection and income security. Non-employment rates for persons with a longstanding illness as well as a low educational level were found particularly high in the Anglo-Saxon and Eastern regions, compared to the Nordic region (17). The poor position of disabled people was suggested to be partly counteracted by employment protection (19), although inequalities in health in itself were not consistently smaller in the Nordic region (20).

Last paragraph introduction changes:

Limited information exists on trends in health-related educational inequalities in paid employment among persons in different European regions, since most studies have focused on a single country or compared a selected group of countries. Therefore, the aims of this paper are (i) to examine absolute and relative differences in paid employment between persons with and without a chronic illness within each educational group in five European regions and (ii) to describe trends in socioeconomic differences in paid employment between people with and without a chronic illness within European countries.

Comment #2:

Indeed, with reference to the previous comment (1), the comparison between 2005 and 2014 does not take into account that the dynamics of changes observed may be influenced by economic and political factors. In particular, it is not considered that changes in the political, economic and institutional conditions of each country or groups of countries might have affected the differences over the years. Moreover, the role of different public policies and institutional aspects that can be responsible for these changes is not discussed enough.

Reply to comment #2:

See our reply to comment #1, the aim of the paper to describe the patterns and dynamics of health related inequalities in paid employment over time, since this overview is missing in the literature. By

using repeated cross-sectional surveys the data presents a snapshot of the labour force situation of each region for the respective years. The aim is not to explain the trends by analysing the specific institutional or political factors in these regions/countries. In the discussion we do point towards some underlying mechanisms that could be considered to influence these findings, such as economic differences, but it is outside of the scope of this paper to further analyse these mechanisms.

We have added a reference in the last paragraph of the discussion that refers to the severity of the economic crisis and the differences between the regions. The beginning of the paragraph has been amended: "The trend analysis might be influenced by the differential effect of the economic crisis across regions." We have added some lines regarding the trends in educational inequalities in health ("Furthermore, health may have worsened during the economic crisis. For persons with existing health issues, this could have led to more difficulties maintaining paid employment. A recent study found an increase in absolute inequalities in health between educational groups after the recent economic crisis in the Continental and Anglo-Saxon region (41). This may explain the increasing trends we found in these regions.").

Furthermore, we rewrote the lines about future research in the discussion so we provide some paths for further inquiry in future research as you suggested.

Old sentence: The increase in precarious employment may be partly responsible for increasing educational inequalities in labour force participation, but this particular path of inquiry needs to be studied further.

New sentence: Further research is necessary to evaluate whether the increase in precarious employment could be an explanation for increasing educational inequalities in paid employment.

Comment #3:

In order to better frame the topic, a link with the existing literature and the contribution of the paper should be clarified. References from the scientific literature should be strengthened.

Reply to comment #3: The references to the scientific literature focusing on socio-economic inequalities in health in Europe over time and the association between labour force participation and health have been added (ref. 7 (Schuring et al. 2015) and 41 (Leao et al. 2018)).

Comment #4:

The modelling strategy is not clearly motivated and described. The Authors should better explain how the Cox regression overcomes the main limits of the traditional logistic regression in their specific field of study. In other words, I think the advantages that could derive by the use of Cox regression instead of the classical logit/probit models are not too much emphasised.

Reply to comment #4: A sentence has been added to the methods section stressing that the Cox proportional model will more accurately estimate the relative risk than the odds ratio.

Added line in the statistical method section:

"This model will more accurately describe the relative risk between being in paid employment and having a chronic illness than an odds ratio, since the odds ratio will overestimate the prevalence ratio due to the high prevalence of being in paid employment (24, 25)".

Comment #5:

Moreover, some additional lines should be devoted to explain how potential problems of sample selection regarding the dichotomy being/not being in the labour force and the way through which it

could be treated. I should stress a little bit more the crucial point of the proposal that is the use of Cox regression.

Reply to comment #5:

The EU SILC data uses complex sampling schemes that differ across countries, and therefore not a random sample of the population where all persons have an equal probability of being included. This design can affect the use of statistical techniques such as logistic regression. For instance, if the sample is not representative, weights should be utilised in order to calculate correct confidence intervals in logistic regression. Relative differences calculated with the Cox proportional hazard model are less influenced by the sample design. If for instance less people with a chronic illness participate, this will not affect the relative difference.

Added lines in Strengths and Limitations: "By utilizing Cox regression instead of logistic regression the different sample designs in EU SILC across countries will not affect the results of the statistical analysis."

Comment #6:

In the discussions, some interesting food for thought for future research is provided. The results, though, imply policy implications of which no suggestions are given or they are very poor.

Reply to comment #6: The focus of the paper is on whether people with health issues are in paid employment, whether it is socially patterned and whether there have been changes in the period 2005-2014. A good overview of health-related educational inequalities in paid employment on a European scale is missing in the literature. This comment links to comment #1 of the first reviewer, who also considered the policy implications to be out of sync with the aim of the paper. Therefore, we have removed the reference in the policy implications to policies and programmes to improve the position of those with health issues, which have not been analysed in this study.

VERSION 2 – REVIEW

REVIEWER	Punzo, Gennaro Univ Naples Parthenope, Italy
REVIEW RETURNED	04-Oct-2018

GENERAL COMMENTS	Overall, the paper has greatly improved, because the Authors seem to have followed the recommendations that were made by the two referees. However, at a second reading, failure to reach a decent scientific quality is made apparent. My first main concern is that, even if it has improved a lot, is not yet accurate enough and should be greatly improved. My second main concern is that the statistical model is not yet justified appropriately. The modelling strategy is not motivated enough while no discussions are on how potential problems of sample selection regarding the dichotomy being/not being in the labour force are treated. In other words, how did the Authors deal with the sample selection bias? Which type (if any) of correction? Which procedure was applied to take into account the different dynamics of participation in the labour market between the genders? Those different dynamics inevitably reflect on the potential differences in behaviours between working and non-working women. See for example: Heckman, J.J. (1979). Sample Selection Bias as a Specification Error". Econometrica. 47 (1) pp. 153–161). In brief, I
---

	think the paper still requires major revisions in order to turn it into an informative account.
--	---

VERSION 2 – AUTHOR RESPONSE

Reviewer's Comments to Author:

Reviewer: 2

Reviewer Name: G Punzo

Institution and Country: Univ Naples Parthenope, Italy

Please state any competing interests or state 'None declared': None declared

Comment #1:

Overall, the paper has greatly improved, because the Authors seem to have followed the recommendations that were made by the two referees. However, at a second reading, failure to reach a decent scientific quality is made apparent.

My first main concern is that, even if it has improved a lot, is not yet accurate enough and should be greatly improved. My second main concern is that the statistical model is not yet justified appropriately. The modelling strategy is not motivated enough while no discussions are on how potential problems of sample selection regarding the dichotomy being/not being in the labour force are treated. In other words, how did the Authors deal with the sample selection bias? Which type (if any) of correction? Which procedure was applied to take into account the different dynamics of participation in the labour market between the genders? Those different dynamics inevitably reflect on the potential differences in behaviours between working and non-working women. See for example: Heckman, J.J. (1979). Sample Selection Bias as a Specification Error". *Econometrica*. 47 (1) pp. 153–161). In brief, I think the paper still requires major revisions in order to turn it into an informative account.

Reply to comment #1: Thank you for your comments regarding the improvements on the paper. Regarding your additional comments, please allow us time to reflect on your thoughts and explain.

1. One of the main concerns of the reviewer is justification of the model and the possible consequences with regard to the results.

In the paper we explain that we use the Cox proportional hazard model with robust variance to calculate relative differences. We have adjusted the text and moved the explanation as to why we selected these models after we presented the models. Text adjustment: "Cox regression analysis was preferred above a logistic regression analysis, since it will more accurately estimate the prevalence ratio, which is much closer to the relative risk between being in paid employment and having a chronic illness, than the odds ratios in a logistic regression analysis, since the odds ratio will overestimate the risk ratio due to the high prevalence of being in paid employment in the current study (21,22)." Reference 21 and 22, as various other papers, explain in detail why this approach is a much better reflection of a relative risk, than in logistic regression analysis that estimates differences in likelihood. We have also clarified in the strengths and limitations the advantage of using Cox regression and have changed the sentence to: "By utilizing Cox regression instead of logistic regression, differences in labour force attachment in EU SILC across countries will not affect the results of the statistical analysis."

For calculating the absolute differences in percentages labour force participation between those with and without a chronic disease within a specific educational category we used linear regression. We

have added information on why we selected linear regression in the method section. Added lines: "In this binary linear regression analysis the dependent variable expresses the proportional difference of being in paid employment (within the range of 0.0 to 1.0), comparing subjects with a chronic disease to subjects without a chronic disease in the same educational group. Thus, the proportional difference equals the absolute difference or gap between proportions of being in paid employment, which is our primary measure of interest (23)."

It is important to state that the aims of our paper were to describe changes in inequalities over time. Our paper was not written to provide estimates of causal mechanisms why those with a chronic disease have lower labour force participation.

2. The second concern is regarding sample selection bias and whether we have corrected for it. Also, particularly the dichotomy being/not being in the labour force.

You refer to the paper by Heckman from 1979 that reflects on issues concerning sample selection. Two types for sample selection bias are brought forward in the paper. One reason is there might be self-selection by participants in the EU SILC surveys. In our case this type of sample selection would mean that the available data and the labour market participation that we see in our sample is not representative of a randomly selected sample of the general population. Although the EU SILC survey is explicitly designed to provide a random, comparable sample across different regions (randomization on household ID), it could be argued that sampling is not completely random and that exactly the same concepts are measured in different cultures. However, other researchers, such as for instance van der Wel et al (ref 27 in paper), have compared the labour market participation data from EU SILC to the EU Labour Force Study and found a high rank-order correlation of 0.80. This comparison provide us with enough evidence to state that the selection is representative of the population.

The second type of sample selection bias could arise due to actions taken by the researcher. In our method section we state that we "selected individuals with complete information on work status, chronic illness, educational background, age, and gender. Our study comprised 179,724 individuals aged 30-59 years in 2014." (Old sentence), however we have not yet mentioned the possible influence of selecting complete cases only. In line with your suggestion, we added the following sentence: "After excluding individuals who were younger than 30 or older than 59 (around 50% of the sample), and deleting observations with relevant missing information (majority due to no information on chronic disease, 16% of the remaining sample), our study comprised 179,724 observations for 2014." Although we did not correct for sample selection we have added to the strengths and limitations section in what way the selection could influence our results: "Sixteen percent of our sample did not respond to the question on chronic illness. These non-respondents had a similar age, were higher educated and more often employed than respondents. Based on their characteristics, non-respondents were less likely to have a chronic illness. Although it is difficult to predict the influence of this selective reporting, the slight underrepresentation of subjects with a chronic disease may have resulted in an underestimation of the association between having a chronic illness and being in paid employment."

Regarding the dichotomy of being or not being in the labour market, in our previous revision comments we mentioned the focus of the paper is whether people with health issues are in paid employment or outside of paid employment. We decided to not focus on disability or unemployment specifically, but to provide a broad overview regardless of the underlying reasons for not being in paid employment. We know that being in poor health is related to unemployment and disability, but that it can also affect the decision to retire early.

3. The reviewer's third concern is whether and how we take into account the different dynamics of participation in the labour market between the genders.

Regarding different dynamics of labour market participation between the genders, we mention in the method section that the analysis has been stratified by gender and region, therefore all models were conducted separately for men and women in each region. All results were separately presented for male and female, because of the differences in labour force participation for gender. In order to clarify why we stratified we have changed the old sentence “The associations were adjusted for age and stratified by region and sex” to “In order to account for differences in labour market participation between men and women across regions [24], the associations were stratified by sex and region, and adjusted for age”.

In the discussion on regional variation in absolute and relative differences between men and women we have changed the sentence “These regional differences between gender may partly be explained by institutional support, i.e. the full-employment policies of the Nordic region, facilitating the integration of women into the labour force by providing for example child care, although often the decision to be active or not is strongly affected by education and fertility choices (37,38).”

VERSION 3 – REVIEW

REVIEWER	Gennaro Punzo DISEG - Università degli Studi di Napoli Parthenope
REVIEW RETURNED	17-Jan-2019

GENERAL COMMENTS	The Authors have implemented most of my suggestions and comments. Overall, the paper is publishable.
--